# Riboflavin and Its Effect on Dentin Bond Strength: Considerations for Clinical Applicability—An In Vitro Study

**DOI:** 10.3390/bioengineering9010034

**Published:** 2022-01-13

**Authors:** Franziska Beck, Nicoleta Ilie

**Affiliations:** Department of Conservative Dentistry and Periodontology, University Hospital, Ludwig Maximilian University (LMU), D-80336 Munich, Germany; nilie@dent.med.uni-muenchen.de

**Keywords:** riboflavin, adhesion to dentin, bond strength, collagen crosslinking, matrix metalloproteinases

## Abstract

Bioactive collagen crosslinkers propose to render the dentin hybrid layer less perceptive to hydrolytic challenge. This study aims to evaluate whether bond strength of dental resin composite to dentin benefits from riboflavin (RB)-sensitized crosslinking when used in a clinically applicable protocol. A total of 300 human dentin specimens were prepared consistent with the requirements for a macro-shear bond test. RB was applied on dentin, either incorporated in the primer (RBp) of a two-step self-etch adhesive or as an aqueous solution (RBs) before applying the adhesive, and blue light from a commercial polymerization device was used for RB photoactivation. Bonding protocol executed according to the manufacturer’s information served as control. Groups (*n* = 20) were tested after 1 week, 1 month, 3 months, 6 months or 1 year immersion times (37 °C, distilled water). The different application methods of RB significantly influenced bond strength (*p* < 0.001) with a medium impact (*η*^2^*_p_* = 0.119). After 1 year immersion, post hoc analysis identified a significant advantage for RB groups compared to RBp (*p* = 0.018), which is attributed to a pH-/solvent-dependent efficiency of RB-sensitized crosslinking, stressing the importance of formulation adjustments. We developed an application protocol for RB-sensitized crosslinking with emphasis on clinical applicability to test its performance against a gold-standard adhesive, and are confident that, with a few adjustments to the application solution, RB-sensitized crosslinking can improve the longevity of adhesive restorations in clinics.

## 1. Introduction

The main reasons for failure of direct resin-based composite restorations are secondary caries and fractures in bulk and margins [1]. A loss in quality, especially in marginal adaption, can be noted as early as in the first three years after placing the filling [2]. The durability of the resin–dentin interface in vivo is impaired by hydrolysis of the interface components, collagen and resin [3]. Due to insufficient infiltration of the demineralized dentin, the collagen fibrils are susceptible to hydrolysis [4,5]. That is proven for both self-etch and etch-and-rinse adhesives [6,7,8]. Deterioration of the hybrid layer can further advance by elution of resin material from the interface [9]. This not only impairs physical properties of the resin–dentin interface [10], but also leaves voids, enabling more water diffusion into the hybrid layer [11] and exposing the previously resin-encapsulated collagen network to proteolytic degradation. Tjäderhane et al. showed that the breakdown of demineralized dentin collagen matrix in carious lesions is closely related to the activity of endogenous matrix metalloproteinases (MMPs) [12]. Furthermore, MMP’s proteolytic activity was also proven in non-carious dentin in absence of bacteria. The associated study further highlighted the dependence of the proteolytic activity of these enzymes on water, as there was no evidence of collagen degradation after 250 days storage in mineral oil [13]. MMPs form a family of zinc-dependent endopeptidases that share characteristics in structure and function [14,15]: Human MMPs generally exhibit a N-terminal signal protein that is eliminated before leaving the cell and linked to a prodomain and a catalytic domain. At the C-terminal end a hemopexin-like domain is found, which is covalently connected to the catalytic domain by a hinge region [15,16]. The prodomain blocks the catalytic center by establishing a close bond between the catalytic zinc ion and the Cys-73 residue of its peptide chain. The disruption of this bond—the “cysteine switch“—displaces Cys-73 as the fourth ligand of the catalytic zinc ion with water which again emphasizes the importance of water for the catalytic action [15,16,17,18,19]. The activation of the zymogen is a stepwise process, mediated by proteinases or non-enzymatic reactive agents such as SH-reactive agents, mercurial compounds, reactive oxygen, denaturants or acidic environments [19,20,21,22]. Mature human odontoblasts and pulp tissue show expression of a majority of MMPs [23], but MMP-2, -8, -9 and -20 are the most common metalloproteinases found in human dentin [24,25,26,27]. The activity of MMPs after adhesive treatment [13] is probably due to zymogen activation through low pH values during interface conditioning. This theory is supported for simplified and two-step etch-and-rinse [28,29] and all kinds of self-etch adhesives [29,30,31,32]. As activated MMPs bear the potential of disintegrating the resin–dentin interface by degrading extra-cellular matrix components, supporting dental caries progression [12,13] and might subsequently lead to complete loss of retention [33], dental research placed a lot of resources in how to stop proteolytic hydrolysis [34,35,36,37,38]. Mainly, two strategies have been pursued: (1) inhibition of proteolytic enzymes, via chelation or blockage of the catalytic center; (2) reinforcement of the collagen network by crosslinkage, rendering it less assailable to proteolytic attacks. In this study we concentrated on the latter, focusing on a crosslinking agent that stands out due to its operating principle.

Riboflavin (RB)—probably better known as vitamin B2—is a non-toxic additive frequently used in the food industry as a food dye or for vitamin supplementation [39]. While in the past the demand of RB was primarily met by chemical synthesis, biosynthesis utilizing fermentation processes and different strains of microorganisms, e.g., Ashybya gossypii, is progressively dominating the field [39]. RB is also a well-researched chromophore known for its capacity to generate reactive oxygen species (ROS) when excited by light [40,41]. This characteristic of RB, combined with the knowledge of possible induction of collagen crosslinks by irradiation [42], has been used in ophthalmology for the treatment of keratoconus [43,44], keratectasia [45] and keratitis [46]. The so-called CXL method could prove to induce: (1) an increase of mechanical rigidity to the collagen network of the cornea [47], (2) an increase in collagen fiber diameter [48], (3) improving resistance against enzymatic collagen degradation [49], and (4) an amelioration in thermomechanical and hydration behavior of the cornea collagen network [50,51,52]. Cova et al. transferred this technique to improve the dentin bonding procedure. They showed increased immediate bond strength for the RB/ultraviolet A-irradiated-group (RB/UVA), and further, the RB/UVA-treatment caused inhibition of selected MMPs [53]. As RB has absorption maxima in both UVA and visible blue light range [54,55], Fawzy et al. introduced photoactivation with a blue light (BL) dental curing unit to the crosslinkage procedure with RB (RB/BL), due to its easier applicability in clinical daily life [56]. Even though RB/UVA activation prove to be more efficient, the collagen matrix after RB/BL still exhibited enhanced mechanical properties and reduced risk of enzymatic degradation as well as improving and preserving dentin bond strength and the integrity of the hybrid layer [56].

The aim of our study was to develop a simplified, clinically applicable and efficient protocol for the usage of photoactivated RB in the dentin bonding process. Therefore, we combined a self-etch adhesive system with RB/BL photoactivation either in administering RB as aqueous solution before the bonding process or incorporating it in the primer. The concentration for the testing solutions was chosen after the study results of Daood et al., who successfully incorporated RB in the primer of an etch-and-rinse adhesive system [57]. Conversely, we chose Clearfil SE Bond 2 (Kuraray Noritake, Chiyoda-ku, Tokyo, Japan) as the adhesive system, as its mild self-etch formulation containing 10-MDP as functional monomer guarantees desirable dentin bond duration and is therefore recommended for dentin bonding by Van Meerbeek et al. [58,59,60,61]. As far as we are informed, a direct comparison of both primer incorporation and pre-adhesive application of RB/BL in a strictly clinics-oriented protocol has not yet been conducted. In addition, the long observation period without artificial accelerated aging is innovative in that context. The efficiency of the different bonding protocols was assessed in a shear bond strength (SBS) test.

Our null hypotheses state that: (1) the usage of RB/BL treatment would not affect neither the immediate nor the 1-year bond strength, and (2) there would be no difference in bond strength values between the different application modes.

## 2. Materials and Methods

### 2.1. Ethical Approval

This study did not include any experiments involving human participants or animals performed by any of the authors. In relation to ethical guidelines, the human teeth used represent residual biological material. For this kind of study there is no consultation obligation by the institutional ethics committee, as stated in the § 24, 2 medical products law. The study was approved under the project number 19-535 KB.

### 2.2. Preparations of Test Solutions

The benefit of using the crosslinking agent RB was analyzed dependent on: (1) the application mode, or (2) the duration of immersion. RB was either applied during the bonding process being incorporated in the primer of a commonly used self-etch adhesive (Clearfil SE Bond 2, Kuraray Noritake, Chiyoda-ku, Tokyo, Japan, LOT 000031) (Table 1) or before application of the primer being used as an aqueous solution. RB was obtained from Carl Roth, Karlsruhe, Germany at a purity level of ≥97% (LOT 026238843). For both testing solutions RB was solved at 3% (wt/vol) either in the primer of Clearfil SE Bond 2 (test group: RBp) or in distilled water (test group: RBs). The testing solutions were kept in UV/light-protective bottles, stored at 4 °C and used within 24 h after being mixed. For the control group no additive was added to the bonding process. Each test group (control, RBp, RBs) was tested after 1 week, 1 month, 3 months, 6 months or 1 year of immersion duration (distilled water, 37 °C), that sums up to a total of 15 testing groups with 20 specimens each (Figure 1).

### 2.3. Dentin Substrate Preparation

A total of 51 sound human third molars were collected and stored in sodium azide solution (0.2%) at 20 °C and used within 4 weeks after the beginning of the experiments. The roots were separated from the crown through a cut slightly below the cementoenamel junction. Then, the crown was cut slightly above the tooth equator into two halves, a coronal and a cervical one, exposing the same dentin surface. Furthermore, each half was cut in four parts representing the mesio-buccal, mesio-lingual, disto-buccal, disto-lingual corner and resulting at optimum in eight parts per tooth in total (tooth parts that did not present the needed dentin surface for standardized bonding—due to differing tooth size—were discarded). Each tooth part was embedded in methacrylate resin (Technovit 4004, Kulzer, Hanau, Germany, Liquid LOT R010050, Powder LOT R010031) in a stainless-steel cylinder (diameter = 16 mm) and stored in distilled water at 37 °C for maximum 72 h. The specimens were randomly allocated to their respective test groups. To create a standardized smear layer and flat dentin surface, all specimens were wet ground with 600 grit silicon carbide paper (LECO, St. Joseph, MI, USA) for 30 s at a speed of 200 revolutions per minute (rpm) on a grinding system (Exakt 400 cs, Norderstedt, Germany) and a one-sided adhesive paper was applied on the specimen reducing the bonding area to a circle round of dentin with a diameter of 3.16 mm (bonding area = 7.84 mm^2^) (Figure 2 and Figure 3).

### 2.4. Bonding Procedure

For the RBp test group, the testing primer incorporated with RB at 3% (wt/vol) was applied on the exposed area and worked in with a microbrush for 20 s. Afterwards it was air dried for 5 s and bonding agent was applied, air dried and polymerized for 10 s with a LED-curing-Unit (Bluephase LED, Ivoclar Vivadent, Ellwangen, Germany, 792 mW/cm^2^) as advised by the manufacturer‘s information. Before the polymerization of the bonding agent, a mold with a height of 4 mm was placed on the specimen, exactly exposing the bonding area (diameter = 3.16 mm) and standardizing the position of the curing unit. For the RBs test groups, the testing solution including 3% (wt/vol) RB was worked in for 20 s and gently dried with air for 5 s, then the unaltered primer and bonding agent were applied and polymerized as described before. Bonding protocol executed according to the manufacturer’s information served as control. A low-shrinkage ORMOCER (organically modified ceramics)-bulk-fill resin composite (RC) (Admira Fusion xtra, VOCO, Cuxhaven, Germany, LOT 1537600) (Table 1) was used as a restoration material, applied into the mold and condensed to a height of 3 mm. Admira Fusion xtra was applied in one bulk as recommended by the manufacturer allowing for an increment thickness of up to 4 mm. The RC was then light cured as described in the manufacturer’s information (Bluephase LED, 792/ 1816 mW/cm^2^, 40 s) (Figure 2). Afterwards, the adhesive paper was removed and the specimens were stored in distilled water at 37 °C in a thermal oven (Jouan EU3, INNOVENS Ovens, ThermoFisher Scientific, Waltham, MA, USA). The distilled water was changed regularly every two weeks until the respective testing after 1 week, 1 month, 3 months, 6 months or 1 year. The testing was carried out with a Universal Testing Machine (MCE2000ST, Quicktest Prüfpartner GmbH, Langenfeld, Germany) with a knife-edge chisel device at a crosshead speed of 0.5 mm/min until fracture (Figure 3).

To ensure thorough polymerization, the irradiance of the LED-curing-Unit (Bluephase LED, Ivoclar Vivadent, Schaan, Liechtenstein) was measured before the start of the experiments by means of a spectrophotometer system MARC (Managing Accurate Resin Curing, Bluelight Analytics Inc., Halifax, NS, Canada) at a distance of 4 mm (792 mW/cm^2^, curing of the adhesive through the mold) and 0 mm (1816 mW/cm^2^; curing of the RBC) between light guide and sensor surface, imitating the exposure distance to the bonding area created by the usage of a mold, which simulates a clinical condition.

### 2.5. Fracture Analysis

The fracture mechanism was analyzed with a magnifying glass with 10-times magnification and assigned to the three following groups: Adhesive failure constitutes a fracture pattern that is located exactly between the tooth and resin composite. A mixed fracture shows partly an adhesive failure but proceeds either through one or both substrates (dentin, RC). A cohesive break is indicated when a fracture line runs only in one or both substrates and not along the bonding area.

### 2.6. Statistics

The data were analyzed with a multivariate analysis of variance (ANOVA; general linear model, partial eta-squared statistics) which assessed the influence of the parameter application (solution, primer, control) and the parameter immersion duration (1 week, 1 month, 3 months, 6 months, 1 year) on the SBS value. Furthermore, the data were statistically evaluated with a one-way ANOVA and the Tukey HSD (honestly significant difference) post hoc test. All data were checked for normality by the Kolmogorov–Smirnov test and the Shapiro–Wilk test. Further, all statistical tests were executed at a confidence level of 95% (IBM SPSS, Version 24.0; Armonk, NY, USA). The homogeneity of variances was confirmed using the Levene’s test (*p* = 0.200). After preliminary studies, the final study protocol and sample size was selected. The power of the sample size was calculated and rechecked after 1-week immersion based on mean and standard deviation of the bond strength values for group RBp and control, which resulted in a power of 89.34% for a sample size of 20. The data were further analyzed using Weibull analysis. The Weibull distribution is a common model used to assess the cumulative probability of default P for brittle materials at applied stress:$$P_{f}\left(\sigma_{c}\right)=1-exp{\left[-\left(\frac{\sigma_{c}}{\sigma_{0}}\right)\right]}^{m}$$

In this equation, $\sigma_{c}$ is the measured strength at failure, $\sigma_{0}$ is the charateristic strength which is defined as the strength at which $P_{f}$ equals 0.632 and *m* is the Weibull modulus. The double logarithm of the aforementioned equation results in the following expression:lnln[1(1−P)]=mlnσc−mlnσ0

The Weibull modulus *m* is the upward gradient of the straight-line graph, resulting by plotting lnln[1(1−P)] against lnσ.

A confidence interval (*CI*) for a confidence level of 95% for the Weibull modulus *m* is computed by calculating the standard error *SE*:$$SE=m\;\frac{\sqrt{1-R^{2}}}{\sqrt{R^{2}\;\left(N-2\right)}}$$

In this equation m is the Weibull modulus, $R^{2}$ is the coefficient of determination and *N* is the number of tested specimens.
CI=SE · θ ± m

When the confidence level is defined as 95%, then θ equals 1.96 in normally distributed data.

## 3. Results

### 3.1. Bond Strength Results

The descriptive statistics (mean, SD) and their analysis with one-way ANOVA and post hoc Tukey HSD test are summarized in Table 2. Furthermore, the visualization of the results of Weibull analysis is presented in Figure 4, with more detailed data in Table 3. The influence of the parameters’ application (solution, primer and control) and immersion duration (1 week, 1 month, 3 months, 6 months, 1 year) was assessed in a multivariate analysis. The parameter application showed a significant influence (*p* < 0.001) with a medium impact (*η*^2^*_p_* = 0.119) on SBS. The factor immersion duration (*p* = 0.109) as well as the combination of both parameters (*p* = 0.307) could not indicate significant effects on SBS.

Except for the 3-month immersion duration (*p* = 0.070), statistical analysis confirmed significant impact for the application mode on SBS for all immersion intervals.

After 1 week of water storage, the RBp group had significantly lower SBS values (*p* = 0.021) when compared to control, but not when compared to RBs (*p* = 0.223). In addition, no statistical difference could be shown between RBs and control (*p* = 0.525), though Weibull analysis demonstrated a superiority in reliability for control (*m* = 4.67 ± 0.39) in comparison to both RBp (*m* = 2.81 ± 0.27) and RBs (*m* = 2.54 ± 0.18).

The data analysis for 1-month immersion duration proved significantly higher SBS values for control when compared to RBp (*p* = 0.001) and RBs (*p* = 0.043).

In contrast to this, post hoc analysis could also not reveal significant variances for any combination of the application modes after 3 months.

After 6 months, a significant difference between RBp and control (*p* = 0.004) was confirmed, whereas no statistically significant difference could be shown either between RBp and RBs (*p* = 0.226) or between RBs and control (*p* = 0.207). However, RBs (*m* = 2.69 ± 0.18) provided more reliable results than RBp (*m* = 2.13 ± 0.22) after 6 months of water storage.

Even though RBs gained superior SBS values compared to RBp after 1 year (*p* = 0.018), there was no statistically significant difference between RBs and control when analyzed in a one-way ANOVA and Tukey post hoc test. However, the Weibull analysis proved that the control group (*m* = 4.00 ± 0.25) conducted more reliable outcome after 1 year of immersion time compared to both RBs-1year (*m* = 2.28 ± 0.11) and RBp-1year (*m* = 2.63 ± 0.34).

The influence of immersion duration on SBS was assessed in one-way ANOVAS. While for both RB application modes (RBp: *p* = 0.792; RBs: *p* = 0.187) no significant impact of immersion time on bond strength was generated, the control group showed significant variance between the immersion intervals (*p* = 0.046). The post hoc analysis further revealed that while RBp and RBs showed constant SBS throughout all immersion intervals, with no indication of statistical difference (RBp: *p* = 0.792; RBs: *p* = 0.187), there was significant reduction in SBS between 1 month and 1 year of immersion duration for the control group (*p* = 0.044).

The Weibull modulus *m* describes the reliability of the tested SBS for each test group: it is depicted as the upward gradient of the straight-line graph, resulting by plotting lnln[1(1−P)] against lnσ. Therefore, the higher *m*, the less scattering of the individual measurements and thus the more accurate the measured data represent the SBS of the respective test group.

When comparing the graphical illustrations (Figure 4) between the three test groups for each immersion interval, we notice an analog development pattern for all three test groups: while at 1-week immersion, all test groups present relatively steep graphs in narrow alignment, a distinct and progressive flattening of the graphs’ slope and a broader scattering of the *m*-values may be noted up to 3 months of ageing. Afterwards, for 6-month and 1-year testing, we see a gradual re-increase for the upward gradient in all three groups, suggesting higher reliability for all three test groups in comparison to 3-month immersion.

Within each test group, both mean and characteristic bond strength (Table 2 and Table 3) followed a very similar variation pattern throughout immersion times, though the performance among the test groups was quite different. While control showed their highest value in both mean and characteristic bond strength after 1 month of immersion and then displayed a gradual decline in bond strength up to 1 year, RBs presented steady results up to the 1-month immersion duration, hitting its low point at 3 months and afterwards gradually re-increased in bond strength. The similarity for both characteristic and mean bond strength values may statistically be explained due to the similarity of Weibull moduli development for all three test groups (Figure 4) as explained above.

As *R*^2^ ranged on a high level between 0.76 to 0.99, presenting predominantly values >0.9, the Weibull regression analysis proves to be a good statistical model fit to predict the data development for the chosen experimental set-up (Table 3).

### 3.2. Fractographic Results

For all tested substrates in general, the fracture pattern was mostly adhesive with 66%, while one third (33.7%) of all breaks were classified as mixed, and cohesive breaking played a minor role with an appearance of only 0.3%. When analyzed in the respective application form, RBp showed at least 70% adhesive breaks and no cohesive fracture pattern for all immersion times. For the RB groups also mostly adhesive break (68%), additionally 32% mixed breaks and 1% of cohesive fractures were noted.

When the breaking mechanism for RBs groups was further assessed dependent on their respective immersion duration, only the 1-year group generated less than 50% adhesive breaks (45%). After 1 year, there was also one cohesive fracture stated (5%) and 50% of mixed breaks.

In contrast to the RB incorporated groups, a mixed fracture pattern was predominant for control specimens (53%), adhesive fracture appeared in 47% of all cases, and again no cohesive break could be recognized at any immersion interval. However, fracture analysis revealed mainly adhesive fractures for control—at least >50%—for the immersion duration of 1 week, 3 months and 6 months. However, after 1-month immersion the vast part of the specimen broke in a mixed fracture pattern (85%). In addition, after 1-year immersion time, the share of mixed breaking was >50%—65% to be exact. As general trend, it may be observed that substrates with high SBS values tended to be associated rather with a mixed or cohesive fracture pattern than to show adhesive failure. More illustrative analysis of the development of fractures for the three test groups dependent on immersion times is depicted in Figure 5.

## 4. Discussion

RB is a non-toxic, water-soluble nutrient [62,63,64] found in many foods and beverages [65,66]. Due to its conjugated aromatic structure, the isoalloxazine moiety, RB, behaves as a chromophore when activated with light [66,67]. RB‘s absorption maxima are located in the blue light and UV spectrum at 446, 375, 265 and 220 nm [55]. In contrast to a dye, RB shows a high intersystem crossing quantum yield (ϕISC ~ 0.7) [67] allowing an efficient transition from the RB excited singlet to the excited triplet state [68,69]. This allows RB to act as a photosensitizer showing both Type I and Type II photoactivation mechanisms [41,70]. Type I photooxidation is characterized by direct interaction between excited triplet-state RB (^3^RB*) and the substrate via electron or hydrogen ion transfer due to ^3^RB*’s high oxidation potential (E ~ +1.7 V). In Type II photoreaction, energy is transferred from ^3^RB* to molecular oxygen (triplet state), resulting in the formation of high-energy singlet oxygen [71,72,73,74]. Which mechanism is favored is highly dependent on the oxygen availability [75], solvent [55,76], solvent polarity [55,77], pH [55] and substrate [74] in each reaction. The photoreactivity of RB is widely exploited in multiple medical applications of different fields: In ophthalmology, combined treatment of RB and UVA light has been applied to cure and alleviate the symptoms of keratoconus [43,44], keratectasia [45] and keratitis [46]. In transfusion medicine, the RB/UVA treatment of platelet concentrates showed promising results in reducing pathogens and ensuring safety of blood products [78,79]. Exploiting the disinfecting properties of photoactivated RB due to the formation of ROS, the method was introduced as an alternative photosensitizer to photodynamic therapy in endodontic and periodontic treatment to dental applications [80,81,82]. However, Cova et al. were the first ones to apply the RB/UVA method to improve the stability of dentin bond strength, pursuing the approach of photooxidative collagen crosslinking inspired by the results presented by Wollensak et al. in the treatment of keratoconus [44,53]. The study did not only show advantage for RB/UVA for the stability of dentin bond strength and reduced nanoleakage over a period of 12 months when compared to the untreated control, but RB/UVA also improved immediate bond strength and lowered the activity of MMP-2 and -9 in a zymographic analysis [53]. Combined with subsequent research results [83,84], general statements concerning the operating principle and the effect of photoactivated RB on dentin can be made: RB/UVA enhances dentin mechanical parameters [56] and renders the hybrid layer less prone to nanoleakage [53,84]. In addition, RB/UVA enhances resistance to the enzymatic degradation of collagen, and RB/UVA inactivates MMP-2/-9/-8 and cathepsin-K [53,85,86]. In addition, RB/UVA increases dentin bond strength: The method showed enhanced immediate bond strength as well as raised micro-tensile bond strength (µTBS) values after both accelerated ageing (thermocycling) and hydrolytic challenge (6/12 months in artificial saliva) in comparison to untreated control [53,84]. RB/UVA furthermore induces crosslinks in the dentin collagen network: Research using micro-Raman spectroscopy on RB photoactivated dentin showed shifting in the fingerprint region of amide-I bands and CH_3_/CH_2_ deformation and amide-III bands, indicating changes in the ultrastructure of the collagen network which are in accordance with previously published work about collagen crosslinking in human sclera [83,87]. The crosslinking capacity attributed to RB/UVA is the reason for the abovementioned changes in dentin characteristics, bond strength and the decreased activity of endogenous enzymes: In the photochemical reaction of RB/UVA with collagen, the forming singlet oxygen, substrate radicals or ROS (superoxide anion, hydroxylradical, hydrogenperoxide) [63] interact with amino acids along the collagen chain through oxidation, resulting in the formation of covalent crosslinks. In particular, threonine, hydroxylysine, hydroxyproline, histidine, tryptophane and tyrosine [88,89] are involved, even though the reaction mechanisms differ. Because of their electron-rich double bonds, tyrosine, tryptophane and histidine are good reactants for singlet oxygen oxidation [65], and histidine is even described as the “primary target“ for singlet oxygen in collagen [88,90]. While tyrosine is oxidized via Type I photomechanism at low oxygen concentration resulting in the formation of bityrosine by radical–radical coupling [65,91], photooxidation of tryptophane shows characteristics of both Type I and Type II pathways forming a mixture of flavin-, indole-, and indole–flavin- associated aggregates [92]. Furthermore, the research of McCall et al. indicates a high dependency of efficient collagen crosslinking on singlet oxygen and free carbonyl groups, identifying them as the primary site of crosslink formation [88]. Additional research shows the involvement of proteoglycans in RB/UVA-mediated collagen crosslinking, emphasizing the complexity of the molecular mechanisms [93].

As RB shows absorption maxima in both UV (375 nm) and visible BL (446 nm) spectrum, Fawzy et al. modified the method, exchanging the UV source for a BL dental curing unit [56]. Even though RB/BL showed to be less efficient compared to RB/UVA, it still showed superior values in mechanical properties and bond strength, improved biodegradation resistance and enhanced and preserved the dentin hybrid layer in comparison to untreated control [56]. Easy clinical availability and applicability are the advantages of using BL activation to induce crosslinking. While more energy can be transferred with the usage of UVA light, BL can penetrate tissue better due to the inverse correlation between wavelength and penetration depth [94]. This enables, theoretically, the formation of a thicker and more organized hybrid layer. Moreover, safety concerns have been raised using UVA in clinical daily life [56]. Thus, we chose BL activation in our study protocol, as we wanted to establish a simplified, clinically applicable, time-effective and efficient routine, which can easily be transferred into clinical daily life. Hence, we also decided for an all-in-one activation, where both RB and the bonding were light-activated/-cured simultaneously. Furthermore, we used a self-etch adhesive system and either administered an RB containing aqueous solution before the primer or incorporated RB in the primer to maximally reduce the number of steps needed for RB/BL photoactivation of dentin. The concentration of the testing solutions was chosen according to the study results of Daood et al. [57] who incorporated different concentrations of RB in the primer of an experimental total-etch adhesive. Their study showed that an incorporation of 3% RB in the experimental adhesive attained the highest bond strength values, while there was no significant decrease in the degree of conversion. The results further suggested that when RB concentrations higher than 3% are used, the simultaneous activation of polymerization and RB with visible blue light results in poor degree of conversion and therefore lower µTBS values [57]. Due to the absorption of visible/UV light by RB, increasing RB concentrations cause competition between RB and the adhesive’s photoinitiator system for photons, resulting in inadequate polymerization of the adhesive layer [56]. In our study we wanted to assess whether the economization of the bonding protocol by incorporating RB in the primer reveals advantages when compared to applying an RB containing aqueous solution before the primer. The first null hypothesis must be rejected, as the RB/BL treatment did indeed affect both immediate and 1-year bond strength: For immediate bond strength, the control group showed significantly superior SBS values when compared to RBp. While there was no statistically significant difference between RBs and control after 1 week, control displayed less scattered bond strength values in comparison to RBs. In addition, the control group provided a more reliable outcome compared to both RB-dependent application modes after 1-year immersion duration. These study results are unlike the research that has been conducted before, as they suggest that the RB-sensitized photoactivation may show adverse effects on dentin bond strength dependent on the formulation of the RB-containing solution. The discrepancy to the study results of Fawzy et al. [56,83] and Cova et al. [53] are easily explicable due to the divergent choices in the study set-up for the photoactivation source, the application protocol and the irradiation time. As their approaches documented the benefits of RB photoactivation in general, our approach was to evaluate whether the dentin adhesive bond could profit from simplified RB-sensitized bonding protocols. Time-consuming, complicated application protocols are less attractive for the dental practitioner as the risk of contamination and thus loss in bond strength increases [95]. More interesting is the comparison with the research of Daood et al. [57], as the study set-up is very similar. The main difference lies in the choice of the adhesive system, while Daood et al. [57] opted for an etch-and-rinse adhesive, we were the first ones to combine RB/BL photoactivation with a modern self-etch adhesive. With the etch-and-rinse approach, the dissolution of the smear layer by etching is separated from the resin infiltration; however, those steps are combined in the self-etch approach, which results in incorporation of smear layer residuals and hydroxyapatite crystals within the hybrid layer [96]. In addition, in comparison to etch-and-rinse systematics, self-etch adhesives result in a shallower demineralization of dentin, exposing the dentin collagen network up to ~1 µm in depth [60]. As the smear layer was not dissolved before the application of the RB-containing test solutions, this might have impaired the infiltration into the collagen network, the actual site of action of RB. Moreover, in future studies, it might be interesting to evaluate whether the application sequence of an RBs test solution before or after the self-etching primer significantly influences the effectivity of RB/BL photoactivation on dentin collagen. Although distilled water as a solvent exhibits a slightly demineralizing effect on dentin [97], the application of RBs after the self-etch primer might improve infiltration of the RB test solution into the collagen network for more effective crosslinking. In addition, the difference in acidity of the primer/adhesive might have had an influence, as the pH of Clearfil SE Bond 2 primer is far more acidic (pH < 2.5) than of usual contemporary total-etch adhesives [98]. Ahmad et al. proved a high correlation between the photolysis of RB and the pH of the solution. The reactivity of the excited triplet state of RB is increased in alkaline solution [99]. Furthermore, due to the dependency of redox potentials on pH [100], the photolysis rate of RB is low at pH 5–6, as the redox potentials of RB are lowest at that range [99]. It is also indicated that, at a low pH range, there is interaction between RB excited singlet state and the buffer species, resulting in quenching of the excited singlet state. All this explains why the highest rate of reaction of RB photodegradation is found at pH 4.3 and at pH 10.8 [99]. Unfortunately, the pH value of Daood et al.’s experimental adhesive was not published [57]. However, it appears probable to assume that its pH, as common for total etch systematics, might have been close to pH 4.3, thus causing the photolysis of RB in the study of Daood et al. [57] to have been more efficient, explaining higher bond strength values and crosslinking capacity in comparison to ours. The parameter application had a significant impact (*p* < 0.001) on the bond strength, so the second null hypothesis must also be rejected. After 1 year, RBs exhibited significantly higher bond strength values than RBp, and this could be again attributed to the pH dependency of the RB photodegradation. As the RB test solution had a higher pH than RBp, it therefore triggered a more efficient RB photoreaction as explained above [99].

The control group, expectedly, presented very favorable results: The formulation of the chosen adhesive, Clearfil SE Bond 2, includes the functional monomer 10-metharyloyloxydecyl dihydrogen phosphate (10-MDP), which forms ionic bonds with the calcium ions of the residual hydroxyapatite in the hybrid layer providing additional stability to the adhesive bond [101]: Thus, in the 13-year randomized clinical trial conducted by Peumans et al. no difference could be found between the gold standard SE adhesive, Clearfil SE Bond and the gold standard ER adhesive, Optibond FL [102,103,104].

Nevertheless, the increase in bond strength for the control group, even though not significant, after 1 month of ageing, was quite uncommon. It may be explained by a plasticization effect of the resin composite, which might have led to a reduced brittleness of the material, and hence might have increased the fracture resistance of the adhesive interface [105,106]. The following significant decline in bond strength after 3 months of immersion is in accordance with general findings about bond strength, as DeMunck et al. described bond strength decreases to be common within 90 days of ageing irrespective of the used adhesive [3].

Interestingly, despite the superiority in SBS for immediate and 1-year results for control, RBs as well as RBp provided more constant results over the course of 1 year, while control in comparison exhibited significant loss in SBS between 1 month and 1 year of immersion duration. This could indicate beneficial long-term effects of collagen crosslinking and MMP inactivation by RB-sensitized photoreaction in a clinically applicable bonding protocol. Both mean and characteristic bond strength displayed a uniform and very similar development irrespective of the test group throughout all immersion intervals, though the increase in both mean and characteristic bond strength for RBs after 1 year of non-accelerated ageing also hinted towards advantages for RB/BL in longer term ageing. For future studies, we therefore recommend study set-ups that entail longer observation periods >1 year, as our results suggested a development towards more articulate effects for RB/BL after the immersion duration of 1 year. In addition, the utilization of accelerated ageing protocols, e.g., thermocycling or the incorporation of either bacterial collagenases or chemical cycling (cyclic exchange of immersion solution adjusted to different pH levels), might result in more pronounced effects for RB/BL on dentin bond strength, even for shorter immersion protocols.

As pointed out in the results section, the fractographic analysis suggested a correlation between high bond strengths with mixed or cohesive fracture patterns: 72.3% of all mixed breaks progressed only through resin composite, which could indicate that the restoration material facilitated crack propagation as it constituted a weaker pathway. While both RB groups presented a very similar distribution of breaks, displaying each a share of 82.4% (RBp) or 80.6% (RBs) of all mixed fractures occurring in the resin composite, the same share for the control group lay only at 64.2%, showing merely doubled shares for “break in tooth” (13.2%) or “break in dentin and resin composite” (22.6%) in comparison to the RB groups. Consequently, the striking difference in fracture progression for control group/RB groups hints towards an intrinsic cohesive toughening effect on dentin collagen through RB application in this study. The common point of critique about shear bond tests exhibiting a high percentage of cohesive failures [107] could not be confirmed by this study because only 0.3% of all breaks were classified as cohesive.

Although bond strength testing with ageing protocols correlates reasonably with medium-term retention rates [107], the obtained data should be interpreted with caution and in relation to the used methodology: All bond strength tests, irrespective of the method, show an inhomogeneous distribution of stress at the adhesive interface and are thus highly influenced by experimental conditions [102,107,108].

Even though the presented results foremost emphasized current limitations of RB/BL, they simultaneously identified starting points for method improvement. Further, some parameters, e.g., the fractography results and the development of characteristic bond strength in the RBs group, hint towards beneficial effects for RB/BL in a longer-term experimental set-up. Thus, for future studies we recommend an observation period >1 year, as we believe that the effects of RB/BL will be more articulate with longer immersion times.

From our perspective, we are positive that with certain alterations to the application protocol, such as the development of the solvent solution with adjusted pH, solvent polarity and viscosity [99,109], RB/BL could be the basic mechanism to an innovative cavity priming solution that could be used in advance to any adhesive treatment in clinics.

## 5. Conclusions

Within the general limitations of a laboratory study, RB shows interesting characteristics that could benefit dental research. Even though some parameters might indicate less long-term degradation for RB test groups in comparison to control, the inferior SBS values show that RB-photosensitized crosslinking cannot yet be transferred into clinics. Our results stress the importance of pH adjustments for the solvent solution, as the efficiency of RB photolysis is best at either pH 4.3 or 10.8. Generally, more basic research would be advisable to better understand the chemical and physical processes that trigger collagen crosslinking and amino acid interactions and how to enforce them more efficiently. As RB photodegradation is highly dependent on the applied wavelength, pH and the respective solvent, we would recommend a separate application of an RB-containing solution to render the formation of ^3^RB* and ROS more effective.

## Figures and Tables

**Figure 1 bioengineering-09-00034-f001:**
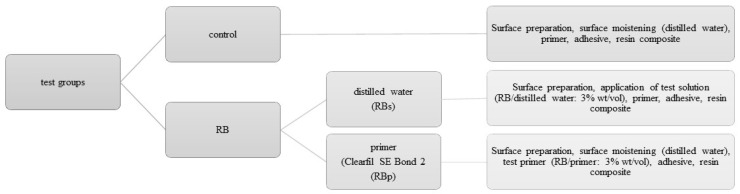
Description of the test groups: each tested after 1 week, 1 month, three months, six months and 1 year.

**Figure 2 bioengineering-09-00034-f002:**
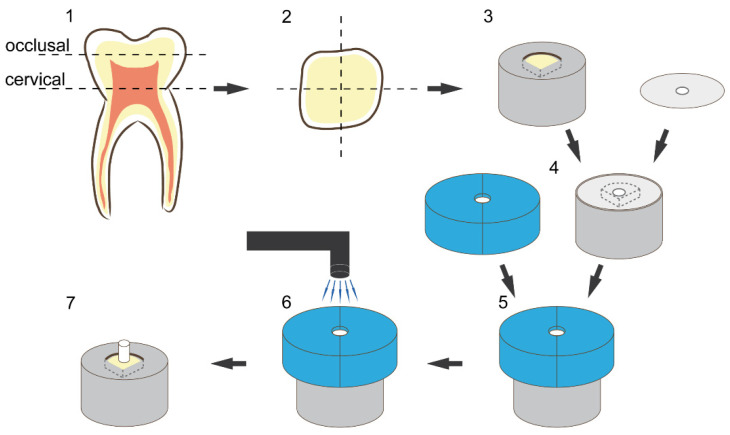
Illustration of sample preparation and bonding procedure: (**1**) schematic presentation of the cutting procedure, (**2**) schematic presentation of the cross-section of each tooth half, (**3**) embedding of the dentin substrate in methacrylic resin, (**4**) placement of the adhesive paper (with a centred circle-round hole; diameter = 3.16 mm) on the specimen to limit the bonding area, (**5**) placement of the vinyl polysiloxane split mold with a cylindric cavity (diameter 3.16 mm, height 4 mm), (**6**) schematic presentation of the resin composite placement and light curing, (**7**) restored final specimen.

**Figure 3 bioengineering-09-00034-f003:**
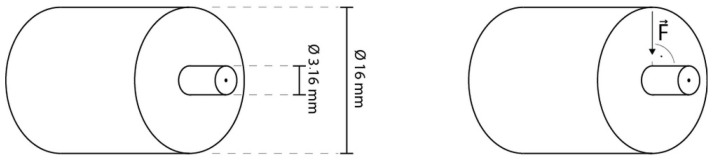
Graphics illustrating specimen dimensions and testing.

**Figure 4 bioengineering-09-00034-f004:**
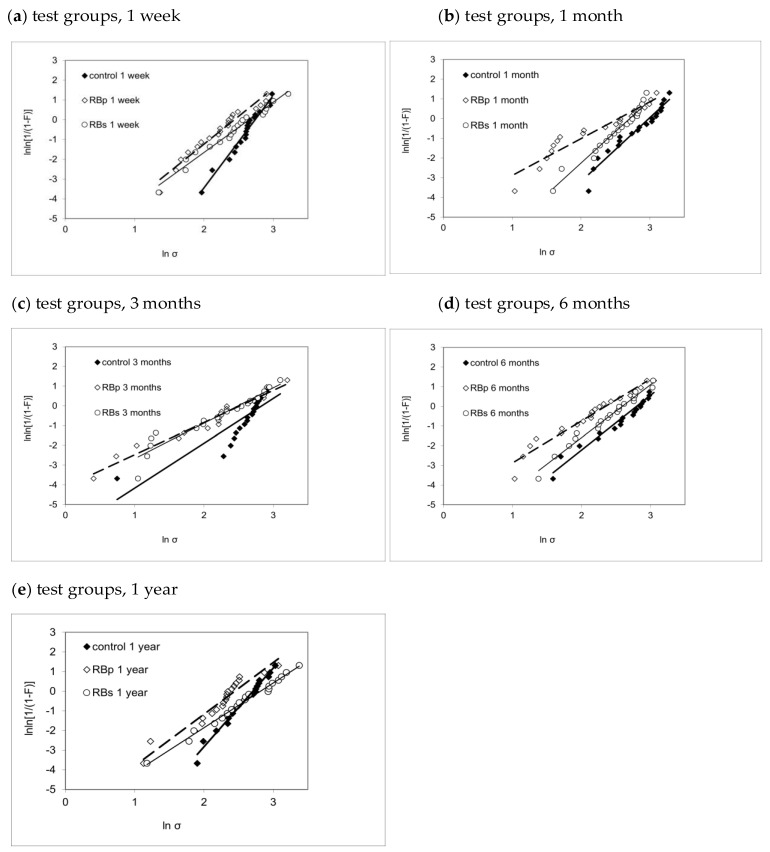
Weibull graphs presenting the Weibull modulus *m*, the associated confidence interval and coefficient of determination for each test group ctrl, RBp and RBs sorted by immersion duration: (**a**) 1 week, (**b**) 1 month, (**c**) 3 months, (**d**) 6 months, (**e**) 1 year.

**Figure 5 bioengineering-09-00034-f005:**
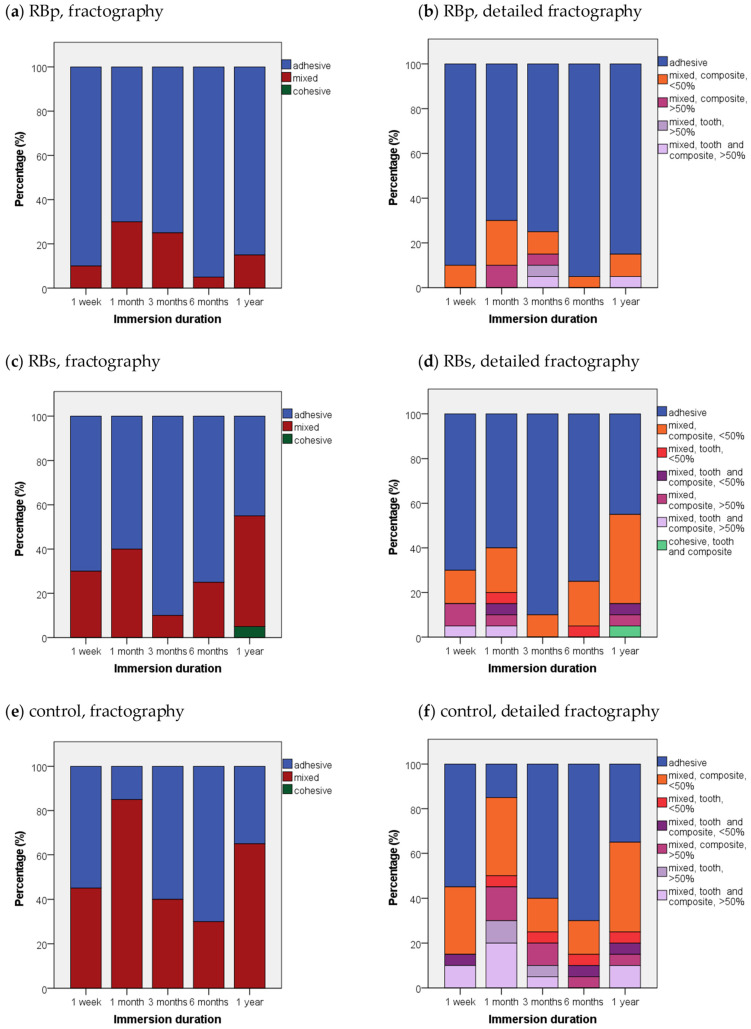
Fractographic failure modes in percent (%) for each test group sorted by immersion times: (**a**,**c**,**e**) representation of simplified fractographic differentiation patterns adhesive, mixed, cohesive for each test group; (**b**,**d**,**f**) additional display of subgroups for mixed and cohesive failure—differentiation between proceeding of break in “tooth”, “composite” or “tooth and composite”; “</>50%” quantifies the coverage of bonding area with substrate in mixed failure.

**Table 1 bioengineering-09-00034-t001:** Characterization and composition of materials.

Commercial Name, Manufacturer, LOT Number	Type of Material	Main Components	Instructions for Use
Clearfil SE Bond 2, Kuraray Noritake, LOT 000031	Two-step self-etch adhesive	Primer: 2-Hydroxyethylmethacrylat 10-Metharyloyloxydecyl-dihydrogenphosphate hydrophilic aliphatic dimethacrylate dl-campherchinone accelerators water dyes	Apply primer to the entire cavity wall for 20 s and dry with mild air for more than 5 s until the PRIMER does not move;
		Adhesive: bisphenol A diglycidylmethacrylate 2-hydroxyethyl metharylate 10-methacryloyloxydecyl dihydrogen phosphate hydrophobic aliphatic dimethacrylate colloidal silica dl-campherquinone initiators accelerators	apply bond to the entire cavity wall and make a uniform bond film using a gentle air flow; light cure bond with a dental curing unit for 10 s
Admira Fusion xtra, VOCO, LOT 1537600	Nanohybrid-ORMOCER bulk-fill resin composite	Matrix: ORMOCER	Apply in ≤4 mm increments; Light cure for 40 s
		Fillers: Based on silicon oxide (84 wt%)	

**Table 2 bioengineering-09-00034-t002:** Mean and standard deviation values (the latter in parentheses) for each test group; the *p*-values represent the results of an ANOVA test for their respective column or row; different uppercase letters signify significant statistical differences in a row; different lowercase letters indicate significant differences in a column—all tested with Tukey post hoc test.

Bond Strength [MPa]	RBp	RBs	Control	ANOVA
1 week	10.20 (4.25) Aa	12.57 (5.50) ABa	14.10 (3.40) Bab	*p* = 0.026
1 month	11.22 (6.27) Aa	13.34 (4.21) Aa	17.66 (5.89) Ba	*p* = 0.002
3 months	10.86 (6.20) Aa	10.61 (5.94) Aa	14.29 (4.14) Aab	*p* = 0.070
6 months	9.24 (4.80) Aa	11.82 (4.82) ABa	14.49 (5.06) Bab	*p* = 0.005
1 year	10.16 (4.19) Aa	14.73 (6.81) Ba	13.58 (3.86) ABb	*p* = 0.018
ANOVA	*p* = 0.792	*p* = 0.187	*p* = 0.046	

**Table 3 bioengineering-09-00034-t003:** Weibull modulus *m*, confidence interval (95%), coefficient of determination *R*^2^ and characteristic bond strength $\sigma_{0}$ calculated for each test group.

Weibull Modulus *m*, Confidence Interval (95%), *R*^2^ and $\sigma_{0}$ (MPa)		1 Week	1 Month	3 Months	6 Months	1 Year
RB 3% (wt/vol)	primer	2.81 (0.27) *R*^2^ = 0.96 $\sigma_{0}$ = 11.44	1.87 (0.25) *R*^2^ = 0.92 $\sigma_{0}$ = 12.71	1.63 (0.15) *R*^2^ = 0.96 $\sigma_{0}$ = 12.42	2.13 (0.22) *R*^2^ = 0.95 $\sigma_{0}$ = 10.44	2.63 (0.34) *R*^2^ = 0.93 $\sigma_{0}$ = 11.52
	solution	2.54 (0.18) *R*^2^ = 0.98 $\sigma_{0}$ = 14.17	3.19 (0.29) *R*^2^ = 0.96 $\sigma_{0}$ = 14.98	1.79 (0.25) *R*^2^ = 0.92 $\sigma_{0}$ = 12.08	2.69 (0.18) *R*^2^ = 0.98 $\sigma_{0}$ = 13.31	2.28 (0.11) *R*^2^ = 0.99 $\sigma_{0}$ = 16.70
Control (no additive)		4.67 (0.39) *R*^2^ = 0.97 $\sigma_{0}$ = 15.43	3.24 (0.36) *R*^2^ = 0.94 $\sigma_{0}$ = 19.76	2.27 (0.30) *R*^2^ = 0.76 $\sigma_{0}$ = 17.02	2.79 (0.28) *R*^2^ = 0.95 $\sigma_{0}$ = 16.42	4.00 (0.25) *R*^2^ = 0.98 $\sigma_{0}$ = 14.98

## Data Availability

The datasets generated during the current study are available from the corresponding author on reasonable request.

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
