# Peer review of "Riboflavin and Its Effect on Dentin Bond Strength: Considerations for Clinical Applicability—An In Vitro Study"

_bioengineering, 2022, doi:10.3390/bioengineering9010034_

Round 1
Reviewer 1 Report
A very well established research, presented in the most appropriate way.
Author Response
Please see attachment. Thank you very much for your commitment.

Reviewer 2 Report
The idea of using RF on dentin is of current interest in dentistry.
This study evaluated bond strength on sound dentin, that is, non demineralized dentin. This means that the eventual effect of RF on bonding improvement is unknown. It is a pity that the authors did not test the hypothesis on demineralized dentin.
The only interesting finding this study is that RF has no adverse effect on dentin, as the differences in bond strength were not significantly different in respect to the control group Table 2, 14.73 and 13.58 after 1 year, . This is clinically relevant as it means that RF can be applied safely on sound dentin and bond strength will not be compromised.
The other interesting finding is that adding RF to a primer is a bad idea and should not be considered in a clinical set up.
Please review the English style, it has to be written in past tense, please modify all throughout the manuscript.
The discussion section and the conclusions should be re written, once again, the samples were non demineralized, so if RF had any effect on collagen, it was masked by the fact that fibers were "protected" by hydroxyapatite.
Author Response
Please see the attachment. Thank you for your commitment.

Reviewer 3 Report
Dear authors,
thank you for the opportunity to review this interesting, very-well written and designed paper dealing with a very novel and relevant topic.
However, there are some points that need to be adapted before publication from my point of view, listed below:
Title:
As your study is an in vitro study, i would make this visible in the title (the title kind of sounds like a review).
Abstract:
L. 11: Though in the abstract, you should be more specific with the kind of specimen right from the begining (e.g. human dentin specimens).
Introduction:
L.28: Although maybe not predimonantly, complete loss of retention should be mentioned as a possible, less examiner-dependent (compared to marginal adaptation) failure of direct composite restorations.
L. 68: If you mention the use of Riboflavin (Ref 38), you should also explain the mechanism in this field shortly.
L. 71: Type I and Type II reactions should either be removed or be explained, if relevant for the manuscript. As they also appear in the discussion, I would remove it from the introduction.
L. 94: Please move LOT-number information into the m&m section.
M&M:
Table 1: You should add more information about the experimental Primer used in group RBp to Table 1. Was it Clear Fill SE primer with RB?
L. 127: Have you obtained written informed consent from the participants?
L. 133: Have you performed completely random dample allocation or was every treatment group present in every tooth?
L. 140: Please add rpm or rcf value for smear layer creation.
L. 160: Please provide working distance for all polymerization procedures (Later, the distance is standardized by the mould).
L. 168: to clearify the abbreviation ORMOCER, i would suggest to write "or", "mo" and "cer" in bold letters.
L.170: Was admira used in increments or in one bulk? Please specify from my point of view, it should be used in incements as supposed by the manufacturer to achieve sufficient polymerization, especially nxt to the adhesive interface).
L. 179-184 is partly redundand compared to L. 160-162. Please adapt.
2.6: I would suggest to move the ethic statement to the beginning of the M&M section.
Results:
3.1 (L.3.1ff.): I would suggest to start with the dexcriptive values followed by results of the statistical analyses.
Table 1, first row: There is a punctation before "Weibull-modulus m" that needs to be removed.
L.276-280: This paragraph presents a discussion of the Weibull modul methodology and should therefore be moved to the discussion section. The same for 294-306. In general, you should divide result and discussion parts more clearly.
Please be very specific with the exact measurement made during the whole manuscript (examples: L. 300: mean bond strangth instead of mean strength, L18 significant advantage regarding bonding strength.) Please double check the whole manuscript regarding linguistic uniformity.
Fig.5: For mor clarity, you should not use the same red shade for different meanings (mixed or mixed composite).
Discussion:
The discussion in general is way too ong. Please make sure that repititions of the introduction are avoided. Some points, as the the van meerbeek quote or the guesswork about the Daood pH-value can be shortened or removed.
L.347-349: Please remove a) b) and c) within sentence for better readability.
L.352: The abbreviation PLT appears here the only time in the manuscript. Please define.
L. 402: To the reader it remains unclear, why Plancks constant is defined here. I would suggest to remove the bracket.
L. 429: What are "relaible" bond strength values? It sounds like a mixture of Weibull and Bond strength, whereas higher bond strength woul be more suitable.
L-489: For this paragraph, a discussion of possible improvements against bacteri-induced stress via cross linked collagen would be interesting (as you used destilled water, which has a demineralizing effect but is not as stressing as organic acids produced by a bacterial biofilm).
Conclusion:
Please keep the conclusion more precise and short. It should summarize your main findings- thus there is no space for discussing other studies (daood) with references in the conclusion.
Author Response
Please see attachment. Thank you for your commitment.

Round 2
Reviewer 2 Report
Thank you for the explanations.
Author Response

(The authors gave the same response as above.)

Reviewer 3 Report
Dear authors,
thank you for your revised manuscript. The paper has improved significantly. From my point of view, there are only few minor points that need adaptation as listed below:
L. 179: Thank you for the explanation regarding the maximum bulk-heigth of Admira fusion. You should add the information that "Admira was applied in one bulk as recommended by the manufacturer allowing a increment thickness of up to 4 mm" to the manuscript.
L. 239: Interpunctation error between "Table 3" and "The"
I
Author Response

(The authors gave the same response as above.)
